# Nanoscale design of polarization in ultrathin ferroelectric heterostructures

Gabriele De Luca [1], Nives Strkalj[1], Sebastian Manz[1], Corinne Bouillet[2], Manfred Fiebig[1] & Morgan Trassin [1]

The success of oxide electronics depends on the ability to design functional properties such as ferroelectricity with atomic accuracy. However, despite tremendous advances in ferroelectric heterostructures, the development towards multilevel architectures with precise layer-by-layer command over the polarization is impeded by the lack of continuous control over the balance of electrostatics, strain, chemistry and film thickness during growth. Moreover, the polarization in the deeper layers becomes inaccessible when these are buried by the ongoing deposition. Taking ferroelectric $BaTiO_3$ and multiferroic $BiFeO_3$ as model systems, we observe and engineer the emergence, orientation and interaction of ferroelectric polarization in ultrathin heterostructures with monolayer accuracy. We achieve this by optical second harmonic generation which tracks the evolution of spontaneous polarization in real time throughout the deposition process. Such direct and in situ access to the polarization during growth leads us to heterostructures with user-defined polarization sequences— towards a new class of functional ferroic materials.

[1] Department of Materials, ETH Zurich, Vladimir-Prelog-Weg 4, 8093 Zurich, Switzerland. [2] Institut de Physique et Chimie des Matériaux de Strasbourg— CNRS UMR 7504, 67034 Strasbourg, France. Correspondence and requests for materials should be addressed to M.T. (email: morgan.trassin@mat.ethz.ch)

I n epitaxial films, control of ferroelectricity now spans from determination of the critical thickness in the ultrathin regime[1, 2] to engineering of ferroelectric properties and domain architectures using epitaxial strain[3–5], surface termination[6], or oxygen partial pressure[7]. Their potential is greatly enhanced by making the step from single films to advanced multilayer architectures[8–12]. Here, the interaction between the individual layers adds to the functionality. In particular, fascinating phenomena happen at or across the interfaces between adjacent constituents[8]. Examples are stabilization of the spontaneous polarization at ultra-low thickness[13, 14], global enhancement of ferroelectricity[15], but also improper ferroelectricity[16], formation of polar vortices[9], and negative capacitance[10, 11]. Implementing the desired degrees of freedom into the multilayer systems remains challenging, however, because of the intricate nature of the interactions among the ferroelectric layers and also between the ferroelectric layers and the paraelectric or metallic spacer layers. In order to achieve deterministic control of the polar state in ferroelectric heterostructures, direct feedback-like access to the polarization of each layer at the unit-cell scale is required. For example, synchrotron X-ray based techniques constitute an advanced approach to in situ investigation of ultrathin ferroelectricity[1, 12], but their application is restricted to highly periodic patterns of ferroelectric domains.

Here we show full access in observation and engineering of the amplitude and orientation of polarization in ferroelectric multilayers for two prominent ferroelectrics, BaTiO₃ and BiFeO₃, using in situ optical second harmonic generation (ISHG) for tracking the evolving polar state. Starting with ultrathin single-layers, the critical thickness and subsequent evolution of the polarization in both compounds are resolved layer by layer. Next, in ferroelectric/metal-spacer multilayers with designed surface termination, we manipulate the relative polarization orientation of adjacent ferroelectric layers. Parallel and opposite orientation of polarization in neighboring layers is initiated and tracked in situ. This level of control opens an avenue towards multilayers with user-defined polarization sequence.

## Results

**Probing the emergence of ferroelectric polarization in situ using SHG.** SHG denotes doubling of the frequency of a light wave in a material. This nonlinear optical technique is sensitive to, in principle, any type of symmetry-breaking long-range order. It has been used to characterize a large variety of ferroic states ex situ[17–20]. It is a non-destructive technique with large working distance. Therefore, it is ideal for in situ probing by connecting a laser and an optical detector to the deposition chamber. In fact, such ISHG has been already discussed as a diagnostic tool, e.g., for the surface chemistry of semiconductors[21, 22]. This is far from the physical electronic functionalities targeted with our work, however. In relation to oxide electronics, ISHG has been suggested as a possible methodical concept, yet without performing any ISHG experiments[23].

In ferroelectrics, the efficiency of the frequency doubling, i.e., the SHG susceptibility, is proportional to $P_s$, the spontaneous

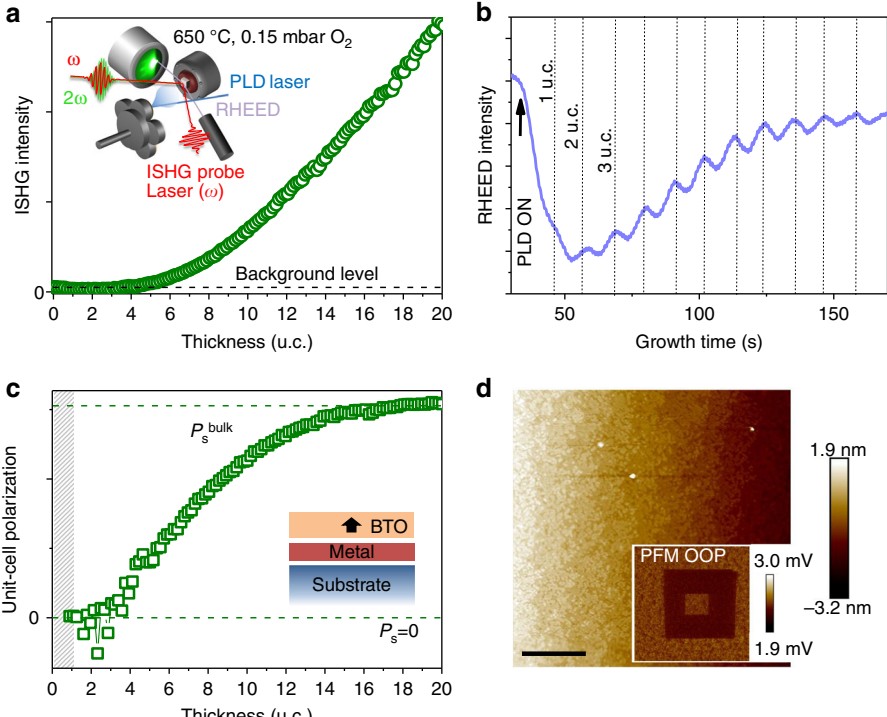

**Fig. 1** Emergence of polarization in ultrathin tetragonal BaTiO₃ films at 650 °C. **a** In situ second harmonic generation (ISHG) intensity dependence of BaTiO₃ (BTO) film thickness during the pulsed laser deposition growth of epitaxial BaTiO₃‖SrRuO₃‖SrTiO₃. The dashed line indicates the intensity of the surface-induced background ISHG signal on the order of 1%. A schematic of the ISHG setup is shown as inset. **b** Reflection high energy electron diffraction (RHEED) measured in parallel to the ISHG yield. **c** Unit-cell (u.c.) polarization as a function of thickness derived by deconvolution from the background ISHG signal (Methods section) and normalization of the square root of the remaining polarization-related ISHG intensity to the film thickness. Data below 1 u.c. are not plotted due to artifacts resulting from the normalization of zero-signal noise. In the inset, the arrow indicates the polarization direction of the BTO single-domain film. **d** Topography and out-of-plane piezoresponse force microscopy scan (PFM OOP) of a BTO film of 30 u.c. that was tip-voltage-poled in a quadratic box-in-box region (see inset). **b**, **d** Do not reveal any evidence for probe-laser damage to the film surface, laser-induced polarization switching or changes in the ferroelectric poling behavior. The scale bar corresponds to 1 µm

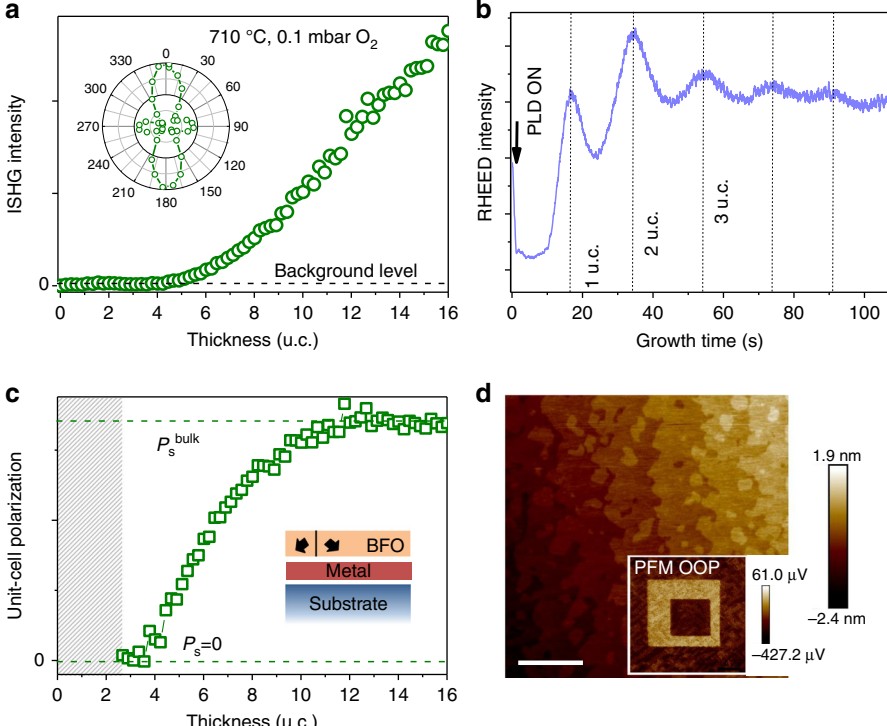

**Fig. 2** Influence of ferroelectric domain pattern on the evolution of polarization. **a** In situ second harmonic generation (ISHG) intensity dependence of BiFeO$_3$ (BFO) film thickness during the pulsed laser deposition growth of epitaxial BiFeO$_3$|SrRuO$_3$||DyScO$_3$ (Methods section). The inset shows the polarization anisotropy of the ISHG yield (varying the polarization of the incident fundamental light and measuring a fixed polarization component at 0° of the emitted ISHG light) at a BFO thickness of 16 unit cells. As explained in the main text, the polarization dependence identifies a ferroelectric stripe-domain structure[19]. **b** Reflection high energy electron diffraction (RHEED) measured in parallel to the ISHG yield. **c** Unit-cell polarization as a function of thickness, derived and plotted as in Fig. 1. **d** Topography and out-of-plane piezoresponse force microscopy (PFM OOP) in the inset of a BFO film of 20 unit cells. The scale bar corresponds to 1 μm

polarization[17, 18]. The sign change resulting from a reversal of $P_s$ corresponds to a 180° shift in the phase of the SHG light wave[24] so that SHG can even distinguish between different directions of polarization in an interference experiment[17], an aspect that will be crucial for the design of polarization-engineered multilayer heterostructures later on. A schematic of our ISHG setup is depicted in Fig. 1a as inset.

In the first step, we need to understand the emergence of polarization in a single film during the growth process with layer-by-layer resolution. We begin with BaTiO$_3$ films that we grow epitaxially on SrRuO$_3$-buffered (001)-oriented SrTiO$_3$ (BTO|SRO||STO, see Methods section for details). Reflection high energy electron diffraction (RHEED) is performed simultaneously to ISHG for calibrating the ISHG yield to the thickness of the BTO film with unit-cell accuracy (Fig. 1b).

At zero coverage, Fig. 1a merely yields a surface-induced background ISHG contribution on the order of 1% of the maximum net ISHG signal. It is deconvoluted from the data (Methods section) before calculating the unit-cell polarization in Fig. 1c. The onset of a macroscopic polarization in BTO, indicated by the steady increase of the ISHG signal, occurs around 4 unit cells. This critical thickness is in agreement with previous reports[1, 25], but here we observe the emergence in a single, continuous experimental run. According to Fig. 1c, the unit-cell polarization of our film saturates around 16 unit cells.

Note that complementary cooling runs on a BTO film of 3 unit cells yielded no polarization-related ISHG signal down to room temperature (Supplementary Fig. 1), we do not observe a temperature dependence in the critical thickness for the onset of ferroelectricity[26]. Figure 1 thus tells us that room temperature

ferroelectricity in heterostructures with BTO layers of <4 unit cells cannot be intrinsic, but must involve some interlayer coupling[12, 13, 27]. Our measurement also shows that tetragonal ferroelectric thin films like BTO can be grown right in the ferroelectric state, in agreement with synchrotron studies[28, 29]. With a growth temperature of 650 °C, $T_C^{film}$ must be substantially higher than $T_C^{bulk} = 120$ °C. Such enhancement is associated with effects like compressive epitaxial strain[28] or strain-induced defect dipoles supporting increased tetragonality[30]. Finally, Fig. 1 points to single-domain growth of the BTO since in a multi-domain film interference of SHG contributions from oppositely polarized domains would lead to cancellation of the ISHG yield[24]. The depolarization fields from surface charges that would promote such nanodomain generation are screened by the metallic SRO buffer layer[31, 32]. The suspected single-domain state is further confirmed by the piezoresponse force microscopy (PFM) measurement in the inset of Fig. 1d. The outer, unpoled region shows the same brightness as the innermost quadratic box that was tip-voltage-poled into a single-domain state.

For scrutinizing the influence of domain formation on the emergence of polarization, we then tested BiFeO$_3$ films grown epitaxially on SrRuO$_3$-buffered (110)-oriented DyScO$_3$ (BFO|SRO||DSO, see Methods section for details). The ISHG data in Fig. 2a, c show that ferroelectricity in the multi-domain films emerges at 4 unit cells. The polarization analysis of the ISHG signal in Fig. 2a, performed as described elsewhere[19], and complementary PFM scans, included as Supplementary Fig. 2, reveal that films develop a ferroelectric stripe-domain pattern[33]. The unit-cell polarization in our BFO|SRO||DSO films saturates at 10 unit cells. We verified that, in contrast, saturation in single-

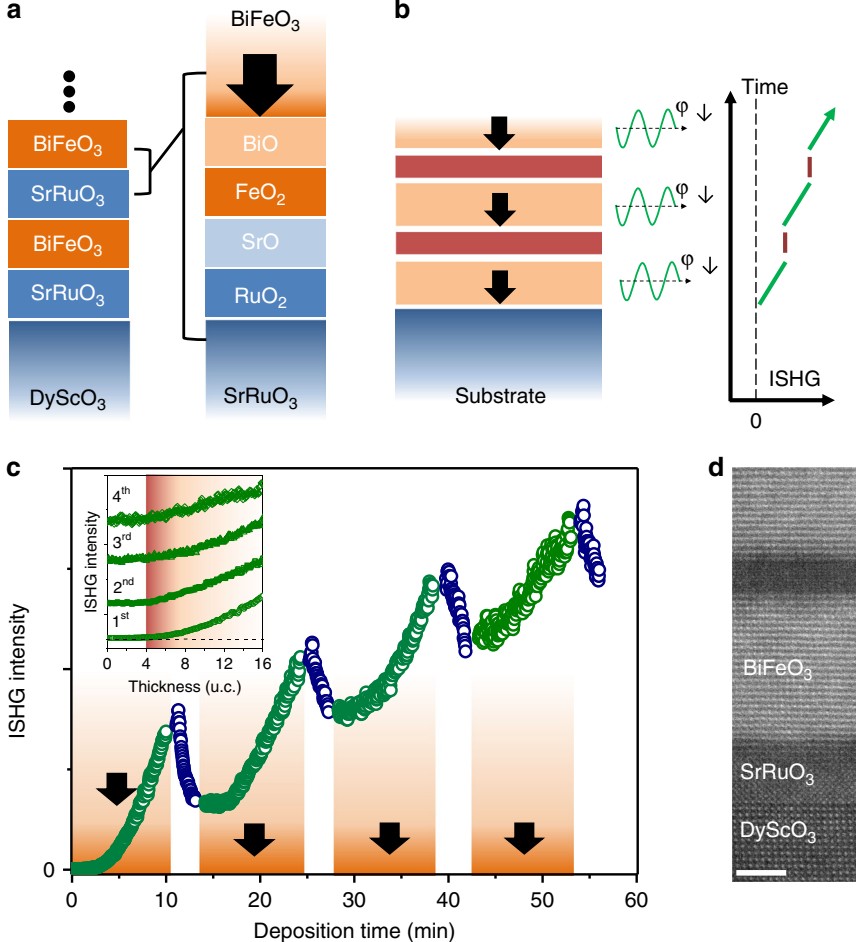

**Fig. 3** Engineering ferroelectric multilayers with user-defined up/down polarization sequence I. **a** B-site growth and downwards polarization of BiFeO$_3$ (BFO) is achieved by SrO self-termination of the SrRuO$_3$ (SRO). **b** Ferroelectric multilayer as in (**a**), assuming a uniform orientation of the polarization in all layers. This should lead to repetitive increase of the in situ second harmonic generation (ISHG) yield because of constructive interference of the ISHG waves emitted from each layer (indicated by the uniform ISHG phase φ$_↓$). **c** ISHG on a sequence of BFO|SRO multilayers. All the BFO layers (20 unit cells, ISHG as green open circles) grow with downward polarization, evidenced by continuous increase of the ISHG intensity from the BFO. During SRO deposition (10 unit cells, ISHG as blue open circles) the signal decreases because of linear absorption. The inset with vertically displaced curves emphasizes the identical critical thickness at which the polar state emerges in all the BFO layers. **d** Scanning transmission electron microscopy image in the high-angle annular dark-field mode confirms the high interfacial quality of the multilayer structure with no visible interdiffusion. The scale bar corresponds to 3 nm

domain (111)-oriented BFO|SRO||STO ferroelectric films grown in parallel is not even reached at 20 unit cells. Most likely, the multi-domain structure of our BFO sample with stripe domains influences the depolarizing field such that bulk polarization in such BFO films is reached at much lower thickness than in the (111)-oriented single-domain specimens. We thus see that information on the domain structure in the ultrathin regime during growth, which would be inaccessible without ISHG, is essential for understanding the early stages in the formation of a spontaneous polarization.

**Monitoring polar states in heterostructures with user-defined polarization sequences**. We now turn towards evolution and control of the spontaneous polarization ±$P_s$ in complex heterostructure architectures. We use the relation between chemical termination and polarization orientation known for BFO single layers[6] and investigate, if this relation remains valid in BFO multilayer heterostructures, or if interlayer coupling effects interfere with or even dominate the influence of termination. For this purpose, we first grew sequences of BFO|SRO bilayers as

shown in Fig. 3a. Here, SrO self-termination of the SRO buffer layer should force $P_s$ in all BFO layers into the down direction (Fig. 3a). As the growth of this structure progresses, the ISHG signal in Fig. 3c adds up layer by layer. (The signal decrease during SRO deposition is caused by transmission and reflection of the fundamental and ISHG waves on the SRO.) This reveals constructive interference of the SHG contributions from each layer and confirms uniform orientation of $P_s$. According to the inset of Fig. 3c, the onset of ISHG occurs around 4 unit cells in all BFO layers. Thus, the BFO layers behave in an independent way, without noticeable coupling between them. In agreement with this, switching of the polarization of the buried layers due to the change of the electrostatic environment with the subsequently deposited BFO layers, which would manifest as a drop of the ISHG yield, is not observed. Despite the complexity of polarization screening processes at polar oxide interfaces[34], this behavior points towards the metallic nature and an efficient screening provided by the 10-unit-cell SRO interlayers[35]. Note that the ISHG approach allows us to probe the buried polarization without the irreversible invasion of our sample that we would have with transmission electron microscopy.

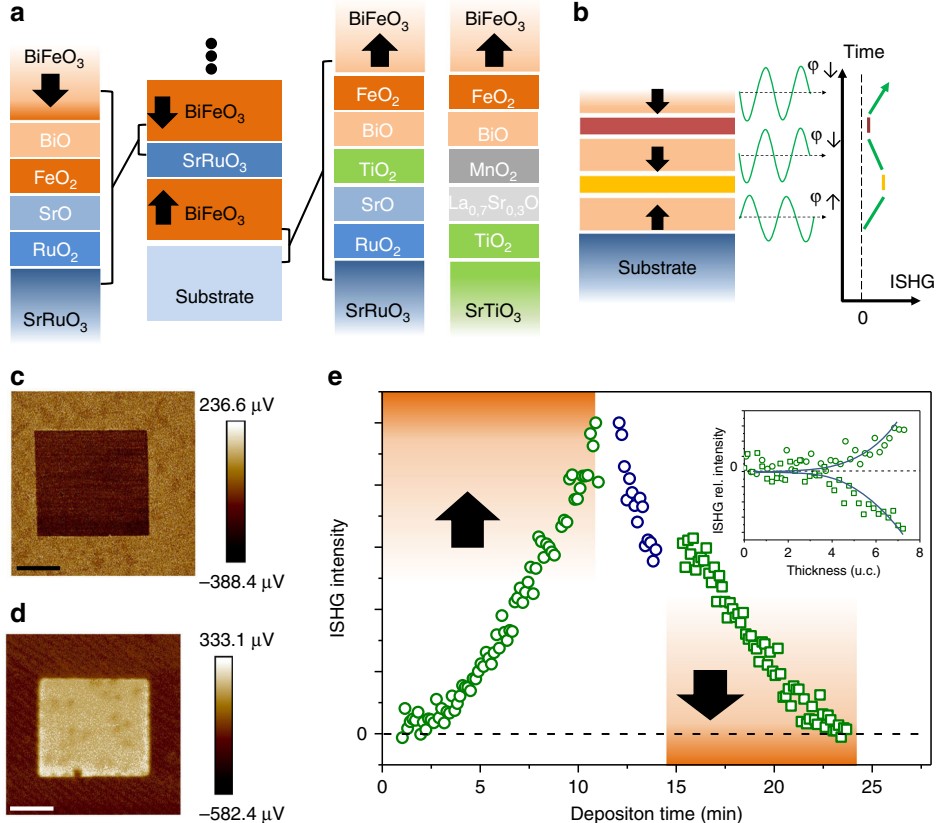

**Fig. 4** Engineering ferroelectric multilayers with user-defined up/down polarization sequence II. **a** A-site growth and upwards polarization of BiFeO₃ (BFO) is achieved by insertion of a TiO₂ monolayer between SrRuO₃ (SRO) and BFO or by substituting SRO by MnO₂-terminated La₀.₇Sr₀.₃MnO₃. B-site growth and downwards polarization of BFO is achieved by SrO self-termination of the SRO. **b** Sketch of multilayer with varying direction of polarization in different layers. The second harmonic generation contributions from these layers may interfere constructively or destructively depending on the respective in situ second harmonic generation (ISHG) phase ($\varphi\downarrow$ or $\varphi\uparrow$ with $\varphi\downarrow - \varphi\uparrow = 180°$). **c** Piezoresponse force microscopy (PFM) scan confirming the up-orientation and reversibility of polarization for an A-site-grown BFO film of 20 unit cells. **d** PFM scan confirming the down-orientation and reversibility of polarization for a B-site-grown BFO film of 20 unit cells. In **c**, **d**, the scale bar corresponds to 1 μm. **e** ISHG from a sequence of oppositely polarized ferroelectric BFO layers (open green circles: up polarization, open green squares: down polarization). The opposite orientation manifests as destructive interference of the ISHG contributions from the two layers. The inset emphasizes the identical critical thickness for the emergence of the polarization in both BFO layers (lines are guides to the eye). Further deposition of down-polarized BFO results in a recovery of the ISHG signal as shown and discussed in Supplementary Fig. 3

The negligible coupling of the BFO layers across the SRO[35, 36] gives us the possibility to set the direction of $P_s$ in each BFO layer independently and thus construct user-defined up–down polarization sequences. $P_s$ is set via the interfacial termination which controls the growth of the BFO from the A-perovskite or B-perovskite ABO₃ site[6]. As mentioned, SrO self-termination of the SRO induces B-site BFO growth with down polarization. TiO₂-terminated surfaces, on the other hand, lead to A-site BFO growth with up polarization. Here we pursue the latter in two ways (Fig. 4b): by depositing a single layer of TiO₂ on the SrO-self-terminated SRO interlayer, or by replacing the SRO interlayer by MnO₂-terminated La₀.₇Sr₀.₃MnO₃-buffered TiO₂-terminated STO[37]. Scanning-probe images in Fig. 4c, d confirm for single-BFO layers that the polarization direction engineering works as expected. This gives us the possibility to orient the polarization independent of the explicit constituents and interface chemistry in a ferroelectric multilayer architecture.

Figure 4e shows the ISHG intensity during growth of a heterostructure with anticipated up and down polarization, respectively, in the first two BFO layers. For the first BFO layer, we observe the onset of ISHG and, thus, polarization around 4 unit cells, followed by further ISHG increase with increasing BFO film thickness. Subsequently, absorption in the SRO layer attenuates the ISHG intensity as explained above. The second

BFO layer, like the first BFO layer, yields an onset of polarization after 4 unit cells, but most strikingly the ISHG signal now decreases with increasing BFO thickness until it finally reaches zero. This clearly points to an opposite direction of $P_s$ in the two BFO layers. The associated 180° phase shift between the $\pm P_s$-related ISHG waves leads to their destructive interference as depicted in Fig. 4a until finally complete cancellation is achieved. Up-polarized and down-polarized ferroelectric layers may, therefore, be stacked at will by using termination control in the metal/ferroelectric multilayers. In particular, there is no reorientation of $P_s$ in the first BFO layer when the SRO or the second BFO layer is grown. Note that only continuous real time observation during growth allows us to conclude that the zero net ISHG signal in Fig. 4e is not caused by an absence of ferroelectricity but by cancellation of opposite ferroelectric polarization contributions in the multilayer stack. This is elaborated further in Supplementary Fig. 3.

## Discussion

In conclusion, we demonstrated how to tailor ferroelectric heterostructures with user-defined polarization sequence. First, we resolved the thickness-dependent evolution of polarization in ferroelectric ultrathin films, using ISHG as non-invasive real time

probe of the polarization throughout the entire growth process. We thus observed the emergence, build-up and bulk-like saturation of the unit-cell polarization in two important ferroelectrics, single-domain $BaTiO_3$ and multiferroic multi-domain $BiFeO_3$. The saturation in the single-domain sample is acquired at higher thickness than in the multi-domain sample. By combining electrostatics, strain and surface chemistry we then show how ferroelectric multilayers with an, in principle, arbitrary sequence of up-polarized and down-polarized ferroelectric layers can be grown.

The unprecedented degree of freedom in tracking and controlling polarization is only one possible route towards a novel class of oxide-electronic devices. Going beyond the ferroelectric properties, ISHG may be employed to track electronic states[38], magnetic-ordering phenomena[17], magnetoelectric-coupling effects[39, 40], and many other functional oxide-heterostructure correlations.

## Methods

**Sample preparation.** The thin films and heterostructures were grown by pulsed laser deposition (PLD). The $DyScO_3$ substrates (Crystec GmbH) were kept at 700 °C during the $SrRuO_3$ deposition and at 710 °C for the $BiFeO_3$ growth. The $La_{0.7}Sr_{0.3}MnO_3$ layer was grown at 700 °C. Oxygen partial pressure was 0.1 mbar in all cases. On $TiO_2$-terminated $SrTiO_3$ substrates (Crystec GmbH), the $BaTiO_3$ growth with SRO as buffer layer was done at 650 °C with oxygen partial pressure 0.15 mbar. The KrF excimer laser intensity was set to 0.9 J cm$^{-2}$. Complementary topography and PFM scans of the BFO films and heterostructures are shown in Supplementary Figs. 2–4.

**In situ second harmonic generation.** SHG is a nonlinear optical process denoting the emission of light at frequency $2\omega$ from a crystal irradiated with light at frequency $\omega$. This is expressed by the equation $P_i(2\omega) = \varepsilon_0 \Sigma_{j,k} \chi_{ijk}^{(2)} E_j(\omega) E_k(\omega)$, where $E_{j,k}(\omega)$ and $P_i(2\omega)$ are the electric-field components of the incident light and of the frequency-doubled polarization, respectively, with the latter acting as the source of the SHG wave. The nonlinear susceptibility $\chi_{ijk}^{(2)}$ characterizes the ferroelectric state[18]. The fundamental light pulses at $\omega$ were emitted at 1 kHz from an amplified Ti:sapphire laser system with an optical parametric amplifier. The light pulses possessed a photon energy of 0.95 eV and a pulse length of 120 fs. To monitor the PLD growth process in real time, we used a custom-designed growth chamber (TSST B.V.) into which we guided the probe light for the ISHG process in a reflection configuration. The incident $\omega$-wave and the emitted $2\omega$-wave were polarized parallel to the reflection plane (p-in/p-out configuration). Laser pulses of about 20 µJ were focused onto a target area with a diameter of 250 µm. The ISHG measurements are performed in real time and without interrupting the growth process. The ISHG acquisition rate was 1 data point per second (1000 laser pulses). Details of the optical setup and the polarization selection rules for SHG on BFO and BTO are given elsewhere[19]. The surface-related ISHG background signal on the order of 1% of the maximum net ISHG yield was deconvoluted from the polarization-induced ISHG signal by describing the measured total ISHG signal as interference between a thickness-dependent polarization-related and a thickness-independent background ISHG wave. The background contribution stems mainly from interface and surface contributions and may possess a phase shift to the polarization-related contribution. Fit parameters were the relative amplitude and phase of the background contribution.

**Data availability.** The data that support the findings of this study are available from the corresponding author upon request.

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

## Acknowledgements

We thank D. Meier for fruitful discussions and J. Nordlander and C. Becher for experimental assistance. We acknowledge funding through the SNSF R'Equip Program (Grant No. 206021-144988). This research was supported by the EU European Research Council (Advanced Grant 694955—INSEETO).

## Author contributions

M.T. and M.F. initiated this work and supervised the research project. G.D.L., N.S. and S.M. performed the ISHG measurements. M.T. performed the thin film growth with G.D.L. and N.S., C.B. conducted the scanning transmission electron microscopy. All authors discussed the results.

## Additional information

**Competing interests:** The authors declare no competing financial interests.

