## [Peer Review File · Nature Communications]

Reviewers' Comments:

Reviewer #1:

Remarks to the Author:

The authors reported in-situ SHG investigations of BaTiO₃ and BiFeO₃ films during their deposition processes, and investigated a real-time formation of ferroelectric polarizations. In particular, they argued that a 5-unit-cell is a critical thickness for the ferroelectric polarization to appear in BaTiO₃. Also, they presented a nano-scale control of ferroelectric polarization in BiFeO₃ by varying the termination of buffer layers. Although the paper deals with important issues using a well-established experimental technique, it is not so clear how noble this work is not only in the technical aspect but also in the scientific perspective.

First, the in-situ SHG technique was introduced more than twenty years ago, and has been adopted to investigate the surface state evolution for semiconductors and oxide thin films. (Yamada & Kumura, *Jpn. J. Appl. Phys.* 34, 1102 (1995); A. Rubano et al., *Appl. Surf. Sci.* 327, 413 (2015).)

Second, authors claimed the critical thickness for the ferroelectric polarization in BaTiO₃ is 5-unit-cell by observing the onset of the SHG intensity. Strictly speaking, however, this claim is valid only at the growth temperature of 650 °C where the SHG characterization was made. With the available results presented in this work, however, one has no information to judge with whether the film is ferroelectric or not at the lower temperature, for example, at room temperature. Actually, it is known that the ferroelectric transition temperature becomes lower as the film gets thinner (in Ref. 13, for example). So, there is an enough possibility to observe the SHG signal at the lower temperature for the thinner films than those presented in this work.

Third, for the BiFeO₃ thin film authors presented a variation of the polarization orientation depending on the termination of buffer layers. However, the basic idea has been demonstrated already by using other experimental approaches (Ref. 7, for example) Although the present work based on the SHG technique can provide clearer information about the ferroelectric polarization, it does not provide additional scientific insight compared to previous studies.

I therefore cannot recommend the publication of this paper as a present form. Here are some other minor issues to be addressed. What is an origin of the oscillation of the ISHG intensity observed in Fig. 2a? The argument will become more strengthened if one can see the increase of the ISHG intensity in Fig. 4e with the deposition of the third BiFeO₃ layer as depicted in Fig. 4a.

Reviewer #2:

Remarks to the Author:

This manuscript reports on a study of ferroelectric properties of BaTiO₃ and BiFeO₃ thin films during growth. This is achieved using second harmonic generation in an in-situ setup where the probing light is introduced in the growth chamber in a reflection mode. Real time SHG signals are analyzed in conjunction with RHEED patterns and the appearance and evolution of electrical polarization with thickness can be precisely determined. Firstly, single compounds are measured and the emergence of polarization as well as its saturation are studied. Then, heterostructures are studied where it is shown that polarizations of the respective BiFeO₃ layers can be controlled to be in a parallel or antiparallel arrangement.

This is a very interesting piece of work showing the power of the SHG technique in this original setup. The measurements clearly demonstrate the onset, evolution and saturation of polarization during growth with an atomic thickness resolution. They also convincingly show the monitoring of polarization direction in the multilayers. These results confirm the extreme control one can achieve when growing ferroelectric heterostructures and the real-time information is an important addition

to previously reported ex-situ analyses of similar systems. The manuscript is clearly written and the data convincingly presented. I think this deserves publication in Nature Communications.

I have a few questions regarding some of the points made in the manuscript which I would like the authors to answer.

1) On the technique itself, the authors refer to their previous recent publication on the subject (ref. 19), but I think some extra information would be useful in the present paper. In particular, in their geometry, interfaces should also contribute to a signal. However, the SHG intensity starts from zero in the measurements. Could the authors explain why this is so? Moreover, one would also expect a contribution from the internal depolarizing field. Could this be observed and if so, could one extract more information from the measurements?

2) For the BTO, the growth temperature at which SHG is measured is way above T_c . The existence of a net polarization at such high temperatures is attributed to strain-induced defect dipoles increasing tetragonality, as shown elsewhere. This is indeed likely but the conclusion that 4 unit cells are required for polarization to appear is a little abrupt as the measuring temperature of 650°C should be underlined. From what is shown in the manuscript, no evidence is provided that polarization does not exist at lower temperature for smaller thicknesses. Did the authors measure ex-situ at room-T with their PFM? If so, this would be worth mentioning.

3) I agree with the conclusion that the multidomain state influences the thickness at which full saturation is observed in BFO. However, I note that it is not shown in any of the figures that the samples have stripe domains. Thus, the statement in figure 2 that 'The polarization dependence identifies a ferroelectric stripe-domain structure' is unsupported as no topography can be inferred from the SHG intensity. Moreover, did the authors not measure the films ex-situ using PFM? Such an image would be useful to strengthen the point.

4) The work on the heterostructures is, in my view, very nice. I am just wondering if the authors also carried out PFM measurements ex-situ? If so, it would be nice to see their results.

Reviewer #3:

Remarks to the Author:

The paper by Trassin and co-workers presents a detailed investigation of ultrathin ferroelectric films and heterostructures, by taking BaTiO₃ (BTO) and BiFeO₃ (BFO) as model systems.

The paper is well written and is put in the relevant literature context. The reported experiments and their discussion are of very high quality and I have no doubt that the paper will raise interest in the community. I am convinced that the paper will be ultimately acceptable and meet the required quality standards of Nature Communications. I recommend publication after minor revisions, which mainly ask for adopting and toning down several parts of the text.

The most original point of the paper is the ability to follow a SHG signal in-situ, during the growth, of functional ferroelectric oxides. Given the characteristics of SHG, this allows characterizing and following the interesting emergence and stabilization of the ferroelectric polarization in ultrathin films and heterostructures. Such SHG observations are outstanding and have to my knowledge no counter-part in literature.

- Having said this, I am puzzled that the authors orient their article already in the title and the abstract mainly around the "nanoscale control of polarization" and the original SHG results seem to be rather a by-product. From my perspective, I think it is a pity that the SHG part is not more commented. How does the technique compare to other techniques? What are the technical difficulties and procedures (more detailed experimental setup, how in-situ are the SHG measurements, truly continuous? Synthesis stopped during measurement? Any other example in literature?). What is the relevance for other materials classes? Magnetic information? All this would even the more allow to appreciate the difficulty, originality and usefulness of the approach. On the

other hand, I believe that the authors overstate at multiple places in the text when they state that they “control the polarization and orientation of spontaneous polarization” by SHG (abstract and similarly throughout the text). Rather than “controlling”, SHG is observing properties, the knowledge of which then allows adjusting synthesis conditions and in turn tailoring properties.

- In the introduction the authors claim “full access and control of the amplitude and orientation of polarization”. Again, the word “control” is an overstatement as it suggests that the authors can introduce whatever orientation and amplitude in the studied systems. For example, how would the authors “control” a random polarization orientation in a monoclinic symmetry?

- In the same direction, I believe that the control of “arbitrary” polarization sequence in heterostructures is also a too strong statement.

- Linking the high T_c in BTO to defect dipoles is likely not the whole story but only one of the ingredients, as stated in the cited original article.

- Figure 1: Please comment on the fluctuation in Figure 1c for thicknesses below 5 uc, what is the origin and why is it not seen in part a? I suggest to have for part a and c the same thickness range

- Figure 2: What is the origin of the oscillations seen in part a and c for thicknesses above 8 uc? Again please adopt the same thickness range in part and c.

- Caption Figure 3 and associated text: I believe that the claim of absence of interlayer coupling is too strong and the inset in part c cannot support this (different curvatures and on-set less clear). The interlayer coupling is small or can be neglected, but its total absence is not proven. What says theory literature on this?

- Conclusion: The first sentence should be toned more carefully. I suddenly see the word confinement, which is not discussed in the text. Confinement is a very specific term and also related to the dimension of the object, I suggest to either discuss this more or to suppress it.

Reviewer #1:

1.1: The authors reported in-situ SHG investigations of BaTiO₃ and BiFeO₃ films during their deposition processes, and investigated a real-time formation of ferroelectric polarizations. In particular, they argued that a 5-unit-cell is a critical thickness for the ferroelectric polarization to appear in BaTiO₃. Also, they presented a nano-scale control of ferroelectric polarization in BiFeO₃ by varying the termination of buffer layers. Although the paper deals with important issues using a well-established experimental technique, it is not so clear how noble this work is not only in the technical aspect but also in the scientific perspective.

First, the in-situ SHG technique was introduced more than twenty years ago, and has been adopted to investigate the surface state evolution for semiconductors and oxide thin films. (Yamada & Kumura, Jpn. J. Appl. Phys. 34, 1102 (1995); A. Rubano et al., Appl. Surf. Sci. 327, 413 (2015).)

We fully agree – but the scope of our manuscript is not to claim the methodical invention of in-situ characterization of thin-film growth by SHG (or other techniques). Rather, our scope is to show that we grow ferroelectric heterostructures with a degree of command over their polarization properties that we can only achieve by applying the (conceptually existing) tool of ISHG. We now realize that it was a mistake to delete a discussion of the ISHG history from an earlier version of our manuscript. We reinserted it on page 3, including the references pointed out by the reviewer. In doing so we point out that Rubano et al. *suggest* the concept of ISHG characterization of oxide thin films – yet *without performing* any ISHG experiments. See also points 3.1 and 3.2.

1.2: Second, authors claimed the critical thickness for the ferroelectric polarization in BaTiO₃ is 5-unit-cell by observing the onset of the SHG intensity. Strictly speaking, however, this claim is valid only at the growth temperature of 650°C where the SHG characterization was made. With the available results presented in this work, however, one has no information to judge with whether the film is ferroelectric or not at the lower temperature, for example, at room temperature. Actually, it is known that the ferroelectric transition temperature becomes lower as the film gets thinner (in Ref. 13, for example). So, there is an enough possibility to observe the SHG signal at the lower temperature for the thinner films than those presented in this work.

The prime goal of our work is to observe the growth and polarization dynamics going on in our samples while it grows at high temperature. But we agree with the reviewer that knowing the resulting room-temperature properties is an important piece of the information. We therefore grew a BaTiO₃ film of three unit cells and tracked its ISHG emission while cooling the sample. Down to room temperature, no SHG emission is obtained. This suggests that the critical thickness observed at 650 °C also holds down to room temperature. Note that even if there *were* a shift of the transition temperature, we could quantify it by interrupting the deposition for an intermittent temperature-dependent ISHG measurement, and then re-heat the sample and continue the deposition. (We found no difference in our samples in comparison to non-interrupted deposition.) We now mention the room-temperature characterization of our samples on page 4 and show the corresponding ISHG data in Supplementary Fig. 1.

1.3: Third, for the BiFeO₃ thin film authors presented a variation of the polarization orientation depending on the termination of buffer layers. However, the basic idea has been demonstrated already by using other experimental approaches (Ref. 7, for example) Although the present work based on the SHG technique can provide clearer information about the ferroelectric polarization, it does not provide additional scientific insight compared to previous studies. I therefore cannot recommend the publication of this paper as a present form.

Ref. 7 discusses termination control of polarization in a *single film*. We show that we can expand the issue of termination control to *multilayer systems*, where effects like interlayer coupling may interfere with (or even dominate) the influence of termination, even if the spacer layer is metallic (see Refs. 8,34). The key to the multilayer systems is ISHG which detects the effect of the deposition on the polarization layer-by-layer and with access even to the buried layers. In our case we see that the BiFeO₃/SrRuO₃ system allows us to grow multilayers with, in principle, arbitrary up/down polarization sequences. We now highlight the importance of the expansion to multilayers on page 5.

1.4: Here are some other minor issues to be addressed. What is an origin of the oscillation of the ISHG intensity observed in Fig. 2a? The argument will become more strengthened if one can see the increase of the ISHG intensity in Fig. 4e with the deposition of the third BiFeO₃ layer as depicted in Fig. 4a.

We identified the oscillation in Fig. 2a as an artifact caused by the mechanical instability of our early ISHG experiments where the laser beam was directed from the neighbouring lab into the PLD chamber. In our recent experiments with the laser next to the PLD chamber, the oscillation never occurred (see Supplementary Fig. 2).

Regarding Fig. 4e, we followed the suggestion of the reviewer and show the deposition of additional down-polarized BiFeO₃ layer to the sample in Fig. 4e. (In a small modification of Fig. 4b, we did this directly, without depositing the additional SrRuO₃ spacer layer.) We observe the increase of ISHG intensity expected by the reviewer. We show this extended deposition as Supplementary Fig 3, along with a detailed explanation of why destructive interference throughout the deposition of down-polarized material first leads to a decrease and then to an increase of the ISHG signal. The reason not simply to replace Fig. 4e by Supplementary Fig. 3 is that this would spoil the very clear impression of destructive interference in Fig. 4e, without adding anything to it, other than a more complex form of ISHG interference. We thus decided to include this more technical aspect as Supplementary Material.

Reviewer #2:

2.1: This manuscript reports on a study of ferroelectric properties of BaTiO₃ and BiFeO₃ thin films during growth. This is achieved using second harmonic generation in an in-situ setup where the probing light is introduced in the growth chamber in a reflection mode. Real time SHG signals are analyzed in conjunction with RHEED patterns and the appearance and evolution of electrical polarization with thickness can be precisely determined. Firstly, single compounds are measured and the emergence of polarization as well as its saturation are studied. Then, heterostructures are studied where it is shown that polarizations of the respective BiFeO₃ layers can be controlled to be in a parallel or antiparallel arrangement.

This is a very interesting piece of work showing the power of the SHG technique in this original setup. The measurements clearly demonstrate the onset, evolution and saturation of polarization during growth with an atomic thickness resolution. They also convincingly show the monitoring of polarization direction in the multilayers. These results confirm the extreme control one can achieve when growing ferroelectric heterostructures and the real-time information is an important addition to previously reported ex-situ analyses of similar systems. The manuscript is clearly written and the data convincingly presented. I think this deserves publication in Nature Communications.

We appreciate this judgement very much, and in particular the underlined statement.

2.2: I have a few questions regarding some of the points made in the manuscript which I would like the authors to answer. 1) On the technique itself, the authors refer to their previous recent publication on the subject (ref. 19), but I think some extra information would be useful in the present paper. In particular, in their geometry, interfaces should also contribute to a signal. However, the SHG intensity starts from zero in the measurements. Could the authors explain why this is so?

Surface or interface contributions are indeed present, but they are on the order of only 1% of the scale shown in Figs. 1 to 4. We now show this more clearly by highlighting the zero ISHG level and the background ISHG level with a tic and a dashed line, respectively, in Figs. 1 and 2. In addition, we properly deconvoluted the polarization-related ISHG signal from the background ISHG for the derivation of the polarization per unit cell in Figs. 1c and 2c and refer to the deconvolution procedure on page 4 and in Methods.

2.3: Moreover, one would also expect a contribution from the internal depolarizing field. Could this be observed and if so, could one extract more information from the measurements?

We agree – depolarization fields are a very important issue and an ongoing project in our group. In the present case we avoid these contributions by the use of metallic buffer and capping layers, and also by the non-uniaxial nature of the monoclinic multidomain-structure of our BiFeO₃ films.

2.4: 2) For the BTO, the growth temperature at which SHG is measured is way above T_c. The existence of a net polarization at such high temperatures is attributed to strain-induced defect dipoles increasing tetragonality, as shown elsewhere. This is indeed likely but the conclusion that 4 unit cells are required for polarization to appear is a little abrupt as the measuring temperature of 650°C should be underlined. From what is shown in the manuscript, no evidence is provided that polarization does not exist at lower temperature for smaller thicknesses. Did the authors measure ex-situ at room-T with their PFM? If so, this would be worth mentioning.

Please see point 1.2 where this issue is addressed.

2.5: 3) I agree with the conclusion that the multidomain state influences the thickness at which full saturation is observed in BFO. However, I note that it is not shown in any of the figures that the samples have stripe domains. Thus, the statement in figure 2 that ‘The polarization dependence identifies a ferroelectric stripe-domain structure’ is unsupported as no topography can be inferred from the SHG intensity. Moreover, did the authors not measure the films ex-situ using PFM? Such an image would be useful to strengthen the point.

The identification of a stripe-domain pattern by SHG is addressed in Adv. Mater. 27, 4871 (2015), and we now refer explicitly to this work. In addition, we performed the suggested PFM measurements and show the expected stripe pattern they reveal as Supplementary Fig. 2.

2.6: 4) The work on the heterostructures is, in my view, very nice. I am just wondering if the authors also carried out PFM measurements ex-situ? If so, it would be nice to see their results.

We performed topography and in-plane PFM measurement. Because of their complementary nature, we show and discuss these measurements in Supplement Fig. 4.

Reviewer #3:

3.1: The paper by Trassin and co-workers presents a detailed investigation of ultrathin ferroelectric films and heterostructures, by taking BaTiO₃ (BTO) and BiFeO₃ (BFO) as model systems.

The paper is well written and is put in the relevant literature context. The reported experiments and their discussion are of very high quality and I have no doubt that the paper will raise interest in the community. I am convinced that the paper will be ultimately acceptable and meet the required quality standards of Nature Communications. I recommend publication after minor revisions, which mainly ask for adopting and toning down several parts of the text.

The most original point of the paper is the ability to follow a SHG signal in-situ, during the growth, of functional ferroelectric oxides. Given the characteristics of SHG, this allows characterizing and following the interesting

emergence and stabilization of the ferroelectric polarization in ultrathin films and heterostructures. Such SHG observations are outstanding and have to my knowledge no counter-part in literature.

We are very happy about this judgement. In particular, we appreciate the underlined statement because it highlights exactly the scope of our work and its relation to the methodical tool of ISHG.

3.2: - Having said this, I am puzzled that the authors orient their article already in the title and the abstract mainly around the “nanoscale control of polarization” and the original SHG results seem to be rather a by-product. From my perspective, I think it is a pity that the SHG part is not more commented. How does the technique compare to other techniques? What are the technical difficulties and procedures (more detailed experimental setup, how in-situ are the SHG measurements, truly continuous? Synthesis stopped during measurement? Any other example in literature?). What is the relevance for other materials classes? Magnetic information? All this would even the more allow to appreciate the difficulty, originality and usefulness of the approach.

After discussion with colleagues, and also based on editorial feedback, we realized that a manuscript restricted to the methodical aspects of ISHG would not reflect our results in a proper way. ISHG as a concept had already been discussed (see in particular point 1.1). In our work we show that by applying the (conceptually existing) characterization tool of ISHG we can grow and understand ferroelectric heterostructures with a degree of command over their polarization properties that we would not achieve without continuous ISHG tracking of the growth process. But we agree with the reviewer that it will be very insightful to include more on the technical aspects of ISHG. We inserted an extended introduction on ISHG on page 3 and in Methods.

3.3: On the other hand, I believe that the authors overstate at multiple places in the text when they state that they “control the polarization and orientation of spontaneous polarization” by SHG (abstract and similarly throughout the text). Rather than “controlling”, SHG is observing properties, the knowledge of which then allows adjusting synthesis conditions and in turn tailoring properties.

We formulated this sentence in a misleading way: Actually, it should be read as: "We control the polarization amplitude and orientation of the entire structure using [= We use] optical second harmonic generation to track the evolution of spontaneous polarization [...]." We checked the entire manuscript and reformulated misleading statements like this, avoiding the impression that SHG controls anything. We also replaced "control" by "design" in the title.

3.4: - In the introduction the authors claim “full access and control of the amplitude and orientation of polarization”. Again, the word “control” is an overstatement as it suggests that the authors can introduce whatever orientation and amplitude in the studied systems. For example, how would the authors “control” a random polarization orientation in a monoclinic symmetry?

- In the same direction, I believe that the control of “arbitrary” polarization sequence in heterostructures is also a too strong statement.

We reformulated statements like this in order to avoid the impression of ubiquitous control.

3.5: - Linking the high T_c in BTO to defect dipoles is likely not the whole story but only one of the ingredients, as stated in the cited original article.

We now mention compressive epitaxial strain and other possible contributions.

3.6: - Figure 1: Please comment on the fluctuation in Figure 1c for thicknesses below 5 uc, what is the origin and why is it not seen in part a? I suggest to have for part a and c the same thickness range

- Figure 2: What is the origin of the oscillations seen in part a and c for thicknesses above 8 uc? Again please adopt the same thickness range in part a and c.

Fig. 1 --- Panel a shows the ISHG intensity, whereas panel c shows the normalization of the extracted polarization (see point 2.2) to the film thickness (= unit cell polarization). In Fig. 1c, we thus divide zero-signal noise by smaller and smaller film thickness which just scales up the noise. The ultra low thickness regime where the normalization results in noise amplification is now marked in dashed area in the corresponding figures. Fig. 2 --- See point 1.4 for the oscillations. For both figures we adopted the thickness range as requested.

3.7: - Caption Figure 3 and associated text: I believe that the claim of absence of interlayer coupling is too strong and the inset in part c cannot support this (different curvatures and on-set less clear). The interlayer coupling is small or can be neglected, but its total absence is not proven. What says theory literature on this?

We softened our statement according to the reviewer's suggestion. We attribute the slope changes in the inset of the figure to a decrease of the growth rate during successive growths.

Regarding literature, at the thickness considered here, the screening provided by SrRuO₃ is expected to be significant according to theory (Nature Mater. 8, 392 (2009)). The effect of incomplete screening and depolarization fields in ferroelectric heterostructures has been investigated in theoretical studies (Rep. Prog. Phys. 79, 076501 (2016)). We added a remark dealing with the complexity of screening mechanism at polar oxide interfaces on page 6.

3.8: - Conclusion: The first sentence should be toned more carefully. I suddenly see the word confinement, which is not discussed in the text. Confinement is a very specific term and also related to the dimension of the object, I suggest to either discuss this more or to suppress it.

We followed the suggestion of the reviewer and refrained from using the term "confinement".

Reviewers' Comments:

Reviewer #1:

Remarks to the Author:

Authors revised the manuscript satisfactorily where I can find overall improvements. However, I have a difficulty in following a logic flow in the introduction part, especially third and fourth paragraphs. Although authors provided an AFM image for the BTO film of three unit cells, I think it is necessary to provide more experimental data demonstrating its high-quality, strained conditions, and so on. With these revisions, I would recommend the publication of this paper.

Reviewer #2:

Remarks to the Author:

The paper by G. De Luca et al. has been revised appropriately. Most of the comments/questions have been answered and I am happy with the new version of the manuscript. I recommend publication in Nature Communications in its present form.

M. Viret

Reviewer #3:

Remarks to the Author:

The reply of the authors to my own and the other reviewers questions and comments looks convincing to me. I much appreciate the clearness of the reply, both in terms of wording and conciseness. The Figures have been much improved.

Concerning more precisely my own remarks, I am totally satisfied by the clarifications and undertaken actions. To my view, the paper can be accepted as is.

Reviewer #1:

1.1: Authors revised the manuscript satisfactorily where I can find overall improvements. However, I have a difficulty in following a logic flow in the introduction part, especially third and fourth paragraphs.

The logic flow is now improved by introducing the “result section”. The description of the experimental setup has been shifted to the end of the paragraph.

1.2: Although authors provided an AFM image for the BTO film of three unit cells, I think it is necessary to provide more experimental data demonstrating its high-quality, strained conditions, and so on. With these revisions, I would recommend the publication of this paper.

We agree with the referee. An AFM image does not guarantee the structural quality of the 3 u. c. thick BTO film. We therefore added the RHEED and diffraction patterns observed during the BTO growth to the data in Supplementary Fig. 1. We also added reciprocal space maps indicating that in-plane strain relaxation only occurs for thick films (more than 30 u. c.).

Reviewer #2, #3:

2.: The paper by G. De Luca et al. has been revised appropriately. Most of the comments/questions have been answered and I am happy with the new version of the manuscript. I recommend publication in Nature Communications in its present form.

3.: The reply of the authors to my own and the other reviewers questions and comments looks convincing to me. I much appreciate the clearness of the reply, both in terms of wording and conciseness. The Figures have been much improved.

Concerning more precisely my own remarks, I am totally satisfied by the clarifications and undertaken actions. To my view, the paper can be accepted as is.

We appreciate this judgment very much.